# Comparison of Real-Time PCR Quantification Methods in the Identification of Poultry Species in Meat Products

**DOI:** 10.3390/foods9081049

**Published:** 2020-08-03

**Authors:** Kerstin Dolch, Sabine Andrée, Fredi Schwägele

**Affiliations:** Department of Safety and Quality of Meat, Max Rubner-Institute, E.-C.-Baumann-Str. 20, 95326 Kulmbach, Germany; kerstin.dolch@mri.bund.de (K.D.); fredischwaegele@gmx.de (F.S.)

**Keywords:** real-time PCR, quantification, chicken, guinea fowl, pheasant, quail, turkey

## Abstract

Poultry meat is consumed worldwide and is prone to food fraud because of large price differences among meat from different poultry species. Precise and sensitive analytical methods are necessary to control poultry meat products. We chose species–specific sequences of the *cytochrome b* gene to develop two multiplex real-time polymerase chain reaction (real-time PCR) systems: one for chicken (*Gallus gallus*), guinea fowl (*Numida meleagris*), and pheasant (*Phasianus colchicus*), and one for quail (*Coturnix japonica*) and turkey (*Meleagris gallopavo*). For each species, added meat could be detected down to 0.5 % *w/w*. No cross reactions were seen. For these two real-time PCR systems, we applied three different quantification methods: (A) with relative standard curves, (B) with matrix-specific multiplication factors, and (C) with an internal DNA reference sequence to normalize and to control inhibition. All three quantification methods had reasonable recovery rates from 43% to 173%. Method B had more accepted recovery rates, i.e., in the range 70–130%, namely 83% compared to 75% for method A or C.

## 1. Introduction

Consumer awareness for food is growing. On one side, this may be due to health, religious, or ideological issues. On the other side, consumers are sensitized due to food fraud incidences like the horsemeat scandal [1]. Therefore, they want to know what they are getting for their money. For processed food products, the easiest information source is the label of ingredients. In the EU, regulation (EU) No 1169/2011 defines specifically what the label should contain and in which order [2]. For their control, if these regulations are complied, affordable and practical analytical methods are necessary. Hence, one of the main focuses of food authenticity testing is to have the right analytical methods in place and, if necessary, to develop new methods or to improve existing ones.

A change in detailedness of analytic results is one point where improvement is needed. Qualitative results are sufficient when many samples are screened to obtain a rough idea with respect to the contamination rate, and to find out suspected cases. However, in processed food with several ingredients, it is not always sufficient to detect a specific ingredient. Quite often, it is more important to know if the content of one ingredient is higher than another one [3], or if the concentration of an ingredient exceeds a certain threshold [4].

To check the correct declaration of ingredients of animal or plant origin, one strategy is to detect a specific sequence of the deoxyribonucleic acid (DNA) of the corresponding ingredient. This is possible as each species has a unique genome. One widely established method for this is the real-time polymerase chain reaction (real-time PCR). It has been used for a long time for different food products, and constantly new methods are published for meat, seafood, milk, and dairy products, as well as fruit juices [5]. The advantages of real-time PCR analysis are its easy handling and affordable laboratory equipment. The biggest issue, however, is to obtain valid results for legal purposes [6]. In literature, there exist several options. 

The most straightforward idea is absolute quantification, where a serial dilution of the target DNA sequence is used as the standard curve [7]. Thereby, the template can be either directly isolated DNA of the pure target [8], a plasmid containing the cloned DNA sequence [9], or synthetically synthesized DNA [10]. This quantification method is well suited for raw food samples. However, for processed food, it is not practicable, as all three DNA sources for the standard curve have in common that they were not treated like the unknown sample. The measurement of DNA is an indirect quantification method for the added amount of animal or plant tissue. The detected concentration of DNA deriving from the target species should correlate directly with the amount of tissue added. But heat treatment may change the amount of DNA detectable due to heat degradation [11,12], which would lead to an underestimation of the actual amount of target tissue added.

The next possibility (method A) is to use a relative standard curve [13]. This step allows to perform quantification of processed and unprocessed food by co-analyzing the DNA of unknown samples together with the DNA of reference material. Therefore, reference material is produced under the same production conditions as the unknown sample. Before production, the target is added in quantities that comply with the measurement range, and DNA is isolated from these DNA standard samples as a reference. For each real-time PCR run, these DNA standard samples have to be applied and measured together with the DNA of the unknown samples [14]. 

Another possibility (method B) is to determine the matrix-specific multiplication factor of each species under each production condition. DNA is isolated from the reference material and from raw meat. These DNA samples are analyzed together to obtain the matrix-specific multiplication factor. In all further quantification experiments, the DNA of the reference material is not needed anymore. Instead, the DNA samples of the raw material are measured together with the DNA of the unknown samples, and corrected with the matrix-specific multiplication factors obtained earlier [3,15,16]. As the matrix-specific multiplication factors vary between laboratories [17], each laboratory has to determine their own multiplication factors for each animal species and each processing condition [17]. To overcome this time- and labor-consuming step, an internal DNA reference sequence is necessary to quantify via a normalized standard curve (method C). This can be a common DNA sequence like *myostatin* or a ribosome subunit [18], which detects the whole amount of eukaryotic DNA. In the subsequent analytical process, either the ratio is determined between target and reference sequence [19,20], or the difference between the detected amount of target and internal reference sequence (ΔCq) [21,22].

This leads to one of the most important decisions of real-time PCR: the choice of the target DNA sequence. While the usage of single-copy DNA sequences is preferred because of the more stable copy number per cell, the application of multi-copy DNA sequences is in favor of lower detection limits [23].

This study focused on the quantification of the relative meat content for five poultry species in meat products as poultry is the most consumed meat, and its consumption rate is still growing [24]. In addition, poultry products are ranked in the top-ten list of most susceptible product categories [25]. The main species for poultry meat are chicken (*Gallus gallus*) and turkey (*Meleagris gallopavo*), while guinea fowl (*Numida melegaris*), quail (*Coturnix japonica*), and pheasant (*Phasianus colchicus*) are less consumed in Germany [26]. For each bird species, a DNA sequence of the mitochondrial *cytochrome b* gene was chosen, which is often used for identifying animal species [27,28]. For the quantification method C, we chose a sequence of the *12S rRNA* gene as it is a mitochondrial DNA sequence as well. To determine if the processing temperature affects the possibility to detect meat from the five poultry species, sausages were prepared under two different temperatures and analyzed with the three different quantification methods.

## 2. Material and Methods 

### 2.1. Material 

#### 2.1.1. Chemical Material

The following chemicals were used: Proteinase K (Machery-Nagel, Düren, Germany), hydrogen chloride, isopropanol, sodium chloride, and tris(hydroxymethyl)aminomethane (Merck, Darmstadt, Germany), dodecyl sulfate sodium salt (Serva, Heidelberg, Germany), ethylenediaminetetraacetic acid disodium salt (Riedel-de Haën, Seelze, Germany), guanidine hydrochloride, DNA-free water, and RNAse A (Sigma, St. Louis, USA), and ethanol (Th. Geyer, Renningen, Germany). The Wizard^®^ Plus Minipreps DNA Purification System was from Promega (Mannheim, Germany), the QuantiTect Multiplex PCR NoRox from Qiagen (Hilden, Germany), and real-time PCR tubes from LTF-Labortechnik (Wasserburg, Germany). 

Primers and probes were synthesized by Eurofins Genomics (Ebersberg, Germany).

#### 2.1.2. Sample Material

All meat samples were pectoral muscle meat. Chicken and turkey meat were obtained from C + C, Kulmbach, Germany, and guinea fowl, pheasant, and quail as whole carcasses from a breeder in Bad Wörishofen, Germany (Josef Maier). All other meat and plant samples were bought in local stores. 

Emulsified type sausages were produced twice as two independent batches A and B on separate days. If necessary, the carcasses were dissected, and the meat was minced in a Bizerba Ladenwolf (Baling, Germany). The basic formulation consisted of 50% meat, 25% sunflower oil, 23% ice, 1.7% nitrite salting mix, and 0.3% phosphate. All % values are (*w/w*) per sausage filling. For each species, the ingredients were added to a meat grinder (Food Machines Saarbrücken MK13, Germany) and mixed at 2600 rpm. The ground meat from all five poultry species were combined in various percentages (Table 1), and then the additional ingredients were added. Sausages were filled into cans (type 99/36 mm, Dosen-Zentrale Züchner GmbH, Cologne, Germany) and cooked at low (75 °C; batch A for 30 min, batch B for 4 min) or high temperatures (117 °C; with final F values of 5.68 and 6.01 for batches A and B, respectively, cf. [29]).

### 2.2. Methods

#### 2.2.1. Bioinformatics

A DNA sequence of the mitochondrial genome was obtained from NCBI GenBank for chicken (NC_040970.1), guinea fowl (NC_034374.1), pheasant (NC_015526.1), quail (NC_003408.1), and turkey (NC_034374.1). These sequences were aligned with the software Molecular Evolutionary Genetics Analysis (MEGA) [30]. Regions with high similarity were chosen for primer and probe binding sites in the area coding for the *12S rRNA* gene. 

The theoretical specificity of all primers was checked with the Primer-BLAST software (Basic Local Alignment Search Tool, NCBI) with the same parameters as described in [14].

#### 2.2.2. DNA Isolation

The Wizard DNA isolation kit from Promega was used as the standard DNA isolation method [31]. All samples were prepared as duplicates according to the corresponding instruction. DNA-free water was used as negative control.

All DNA samples were quantified and qualified by measuring at 260, 280, and 340 nm with a spectrophotometer DU 7400 (Beckman Coulter, Brea, CA, USA). 

#### 2.2.3. Real-Time PCR

##### Reaction Set-Up

All real-time PCR assays were performed on a RotorGene 6000 (Qiagen, Hilden, Germany) according to the QuantiTect Multiplex PCR handbook (Qiagen, Hilden, Germany). The reaction was set up in 25 µL with primer and probe concentrations according to Table 2. The following cycler regime was used: 15 min at 95 °C, 35 cycles of 15 s at 95 °C, and 1 min at 60 °C, collecting the fluorescence signal at the end of each cycle. 

##### Templates

After DNA isolation of the poultry sausages, the duplicates were combined. They were either adjusted to a DNA concentration of 20 ng/µL or diluted 1:10 with elution buffer. The other animal and plant samples were adjusted to a DNA concentration of 2 ng/µL. For the determination of efficiency, R^2^, and limit of detection (LOD) values, the DNA samples of the poultry sausage and of the pure meat samples were diluted ten-fold with elution buffer.

All DNA samples were analyzed in triplicates (standard samples) or duplicates (unknown samples), or in sextets (influence of chicken DNA on detection of pheasant DNA and the LOD). Positive controls and no-template controls (water) were measured once.

#### 2.2.4. Calculation

##### Method A: Quantification with Reference Material

All DNA samples were diluted 1:10 with elution buffer. The DNA samples from S1–5 (Table 1) were used as standard material, and the corresponding Cq values were plotted against the logarithmic starting quantity. This standard curve was used to quantify the amount of meat from each poultry species in the unknown samples U1–5 (Table 1) for both production temperatures.

##### Method B: Quantification with Matrix-Specific Multiplication Factors

For establishing the matrix-specific multiplication factors for each species, DNA was isolated from poultry meat and from the standard emulsified type sausages S1–5 (Table 1) which were adjusted to a DNA-concentration of 20 ng/µL with elution buffer. The DNA from the poultry meat was used as standard material to obtain standard curves. For the detection of chicken and quail meat, this was obtained by using 0.01, 0.1, 1.0, 10, and 100 ng DNA per real-time PCR reaction. For the detection of guinea fowl, pheasant, and turkey meat, 0.1, 1.0, 5.0, 10, and 100 ng DNA per real-time PCR reaction were used. The DNA from the emulsified type sausages S1–5 were used for calculating the respective multiplication factors according to Köppel et al. [13,15] for each meat from poultry species, separately for cooking at low or high temperatures. These multiplication factors were used to calculate the amount of meat from poultry species added in the unknown emulsified type sausages U1-5 (Table 1).

##### Method C: Quantification with Internal Reference Sequence

This method was performed according to Soares et al. [34] with Equation (1):
ΔCq = Cq_target_ − Cq_reference_(1)

The poultry-species-specific real-time PCR systems were used as the target, and the eukaryotic real-time PCR system was used as the reference.

All DNA samples were diluted 1:10 with elution buffer, and the ΔCq values from S1–5 (Table 1) were plotted against the logarithmic starting quantity for the standard curve. With this standard curve, we calculated the amount of DNA in the unknown samples U1–5 (Table 1). This was performed for the meat from each species and under both processing temperatures.

#### 2.2.5. Statistical Analysis

Calculations were performed either with the Rotor-Gene Q Series Software (Qiagen, Hilden, Germany), Excel (Microsoft Office 2019, Redmond, WA, USA), or with JMP (SAS, Heidelberg, Germany). All factors were analyzed by multiple logistic regression, and the chi-squared values were recorded. The level of significance was set at 5%. Standard box plots were used to visualize the data. The box plots show the median, quantiles as boxes, and whiskers extend to 1.5 times the interquartile distance at most. Outliers were not omitted from the analysis.

## 3. Results

### 3.1. Bioinformatics

All primer pairs were checked theoretically for specificity against the ten most commonly eatable bird species (chicken, duck, emu, goose, guinea fowl, ostrich, partridge, pheasant, quail, and turkey). Additionally, the theoretical cross reactivity of the primers was checked for the triplex real-time PCR system (C-G-P) and for the duplex real-time PCR system (Q-T) against all eukaryotes. All false positive matches had several mismatches: the amplicons were either too short or too long, and/or the species were irrelevant as food. Consequently, there were no relevant false positive matches.

The primer pair for detecting all five species was checked theoretically against all entries for animal organisms in the NCBI GenBank database, and amplicons were obtained with a length of 143–146 bp. No mismatches were found for the five poultry species investigated. 

### 3.2. Development of One Triplex and One Duplex Real-Time PCR System

A pentaplex real-time PCR system was proposed in a former publication. However, this system had a lack in precision and accuracy [32]. To overcome this problem, the pentaplex real-time PCR system was split into one triplex real-time PCR system for detecting meat of chicken, guinea fowl, or pheasant (C-G-P), and one duplex real-time PCR system for detecting meat of quail or turkey (Q-T).

DNA isolated from raw meat had concentrations of 333 ng/µL for chicken, 245 ng/µL for guinea fowl, 228 ng/µL for pheasant, 593 ng/µL for quail, and 209 ng/µL for turkey. Ten-fold dilution series of 10^−1^–10^−7^ gave standard curves for detecting the meat of each species. For the triplex real-time PCR system, efficiency and *R*^2^ values were 102% and 0.983 for chicken, 91% and 0.995 for guinea fowl, and 95% and 0.985 for pheasant; for the duplex system, the values were 94% and 0.998 for quail and 93% and 0.993 for turkey, respectively. 

No signal was received with either real-time PCR system when DNA of the following animal species was used: bison, buffalo, camel, chamois, elk, fallow deer, goat, horse, llama, mouflon, pig, reindeer, roe deer, sheep, tuna, wild hare, zebra, or zebu, or DNA from the following plant species: bean, beetroot, black mustard, broccoli, Brussels sprouts, bunching onion, caraway, cardamom, carrot, cauliflower, celery, chili, Chinese cabbage, coriander, cress, cucumber, fennel, garden leek, garden radish, garlic, ginger, green cabbage, horseradish, Indian mustard, kohlrabi, lemon, marjoram, onion, parsley, pepper, pistachio, potato, pumpkin, radish, red cabbage, rutabaga, salsify, savoy cabbage, tomato, white mushroom, white mustard, white pepper, wood garlic, or zucchini.

However, signals were obtained for DNA of chicken, guinea fowl, pheasant, quail, or turkey, each with the respective real-time PCR system (Cq = 15–18) (Table 3). False positive signals were obtained for a few DNA samples with the earliest Cq value of 29. Additionally, all five real-time PCR systems had in common that the blank values gave signals with Cq values between 29 and 33. Therefore, a cut-off was set at Cq ≥ 29.

### 3.3. Quantification

The unknown emulsified type sausages were analyzed with three quantification methods. For each method, the predicted means were calculated for the unknown samples, together with standard deviations, coefficients of variation (CV), and bias. The CV represents the relative standard deviation of results obtained under repeatability conditions, and was accepted with CV ≤ 25 [13].

Bias was accepted in a range of ±25% relative to the mean. Additionally, recovery rates were calculated, and a range of ±30% was accepted, i.e., recovery rates of 70–130% [13].

#### 3.3.1. Method A: Quantification with Reference Material

For this method, DNA of the reference material was used to obtain a standard curve. The reference material with known concentrations of meat from the five poultry species was produced under the same conditions as the unknown emulsified type sausages. Most of the CV and bias values were within the accepted range. Some of the values were out of range, especially when detecting small concentrations of poultry meat or a high concentration of quail meat (Table 4). 

Most recovery rates were within the accepted range of 70–130% (Figure 1). Lower recovery rates (<70%) were obtained for detecting small concentrations (0.5% meat) and higher recovery rates (>130%) for detecting 57.5% quail meat.

#### 3.3.2. Method B: Quantification with Matrix-Specific Multiplication Factors

The DNA of the standard emulsified type sausages were used to calculate the matrix-specific multiplication factors, separately for low or high cooking temperatures, with the DNA of raw meat as standard material. The multiplication factors ranged from 0.90 (for pheasant meat) to 3.82 (for quail meat) at low cooking temperature, and from 0.09 (for pheasant meat) to 0.52 (for turkey meat) at high cooking temperature (Table 5).

All of the CV values were within the accepted range, only the CV value was slightly higher for detecting 0.5% of pheasant meat at low temperature (Table 6). Most of the bias values were as well within the given range, only for detecting chicken, pheasant, and turkey meat some values were out of the range.

Most recovery rates were within the accepted range of 70–130% (Figure 2). Lower recovery rates were obtained for detecting small concentrations of chicken and higher recovery rates were obtained for detecting small concentrations of pheasant meat. 

#### 3.3.3. Method C: Quantification with an Internal Reference Sequence

A mitochondrial reference sequence was chosen because the specific target sequences were mitochondrial. This additional step did not only normalize the results, it also worked well as an amplification and PCR inhibition control, which is recommended for processed food products [13]. For detecting the meat of guinea fowl, quail, and turkey, most of these values are either close to the limit of the range or above (Table 7). On the contrary, the CV and bias values are mostly within the range for detecting chicken or pheasant meat. 

The median of most of the recovery rates were within the accepted range of 70–130% (Figure 3). However, the scattering of the values for the recovery rates were wide for detecting the five poultry meat species. Lower recovery rates were obtained for detecting small concentrations of guinea fowl meat (0.5% meat).

#### 3.3.4. Comparison

Repeatability (CV) and bias were used to compare the three quantification methods. With low cooking temperature, the CV values differed (χ^2^ = 0.0079). For method A, 88% of the CV values were within the limits, 96% for method B, and 67% for method C. For the bias, no obvious difference was seen between method A, B, or C (χ^2^ = 0.3679). With high cooking temperature, the percentage of accepted CV values differed between the three methods (χ^2^ = 0.0395). For method A, 84% of the CV values were within the limits, 100% for method B, and 76% for method C. No differences were seen for the bias values (χ^2^ = 0.4000) (data not shown).

Another criterion is to compare the recovery rates, where the limits for acceptance were set to ±30%. A multiple logistic regression was performed, and all predictors which did not significantly contribute to the whole model (*p*-value > 5%) were removed from analysis. Thus, cooking temperature and batches were omitted from the model. The three quantification methods differed in the percentage of accepted recovery rates (*p* = 0.0110), with 75% for method A, 83% for method B, and 75% for method C. There was no obvious pattern for under- or for overestimation. Recovery rates varied between poultry species (*p* = 0.0129) as well as between concentration levels of poultry meat (*p* < 0.0001). For detecting chicken meat, all three quantification methods had low accepted recovery rates (Table 8). For detecting guinea fowl meat, the quantification method B showed high accepted recovery rates. For detecting pheasant and quail meat, all three quantification methods are similar and well suited. For detecting turkey meat, method A showed the highest accepted recovery rates.

## 4. Discussion

In this study, we compared three different methods to quantify the amount of meat from chicken, guinea fowl, pheasant, quail, or turkey in meat products, cooked at low or high temperatures [24,25,26].

For the detection of the two main poultry meat species—chicken and turkey—there is a large variety of real-time PCR systems. Most of them are single real-time PCR systems to detect a mitochondrial gene like *cytochrome b* [4,23,27,28,35,36]. For the detection of chicken meat, a few chromosomal genes are used like *interleukin-2* gene [37] or *β-actin* gene [38]. Fewer real-time PCR systems have been published for the detection of meat from guinea fowl [39], pheasant [35,39], or quail [35,39]. There is only one multiplex real-time PCR system for the combination of chicken and turkey meat [16]. To our knowledge, less prominent poultry meat species like guinea fowl, pheasant, or quail have not been considered so far. However, these species are also relevant as they are a delicacy and high-priced. Therefore, the focus was set on the combined detection of the two main poultry meat species, chicken and turkey, together with the high-priced poultry meat species guinea fowl, pheasant, and quail.

The bioinformatic testing of the primers resulted in no false positive matches as single systems, as well as within their combinations (C-G-P and Q-T). However, as the DNA databases are not complete, there is always the chance to miss a species [14]. Therefore, different animal and plant DNAs were tested with both multiplex systems. False positive signals were obtained with a few of these DNA samples. However, each of these Cq values appeared later than the Cq values of the blank samples, and each was below the cut-off value. Furthermore, no influence of chicken DNA on the Cq value of detecting pheasant DNA was shown and it was not possible to establish values for the LOD as all dilutions were to 100% detectable until the cut-off. This, together with high efficiency and R^2^ values, indicates that both multiplex real-time PCR systems are precise, specific, sensitive, and suitable to differentiate meat from these five poultry species in meat products.

The two real-time PCR systems were established with DNA that was isolated from 300 mg fresh meat from each species. The DNA content for quail meat was almost twice as high than for each of the other four species. One explanation for this observation is the small size of quails, which is the smallest of the five poultry species investigated. A positive correlation between cell size and body mass among birds [40] implies higher DNA content per body weight in smaller than in larger bird species.

In the literature, there is a large number of methods to quantify material of animal or plant origin in food products [13,41]. Some of these methods are not suited for processed food products, and were therefore not considered in this study. All other methods have in common that standard reference material is required which should be prepared under identical conditions, with similar content, and in similar concentrations [42,43]. Therefore, standard and unknown emulsified type sausages were prepared under comparable conditions.

The three quantification methods compared in this study are in wide use. Quantification method A used DNA from reference material to establish a standard curve that was applied to quantify the amount of meat of each bird species in the unknown samples. At low cooking temperature, the recovery rates were between 70% (chicken or turkey meat) and 90% (pheasant meat) within the accepted limits. At high cooking temperature, the recovery rates were lower than at low temperature for most species, but 100% for turkey meat. Combined with the high bias values for the detection of a low concentration of 0.5% pheasant meat, and 57.5% of quail meat, it can be concluded that quantification with reference material at high cooking conditions is not suited for the whole concentration range for all poultry species. Overall, the idea of this quantification method is quite straight forward, but the main problem is to have the right standard material in stock. For research purposes, this is feasible, and this method was successfully applied to our unknown emulsified type sausages using the standard emulsified type sausages.

Quantification method B applied multiplication factors. This method was first published by Köppel and colleagues in 2011 to detect cow, pig, horse, and sheep [15]. It has been applied to many different animal species since [3,17]. In this study, the multiplication factors were established separately for each bird species and each cooking temperature. The multiplication factors were smaller for high than for low cooking temperatures. This might be due to the higher degradation rate of the DNA due to the higher processing temperature [12]. At low cooking temperature, the percentage of recovery rate values within the limits of ±30% reached from 73% for chicken or pheasant meat to 97% for guinea fowl or quail meat. At high cooking temperature, the percentage of recovery rate values within the limits ranged from 67% (chicken or turkey meat) to 97% (quail meat). Only for the detection of a concentration of 0.5% of pheasant meat, the CV and the bias values were out of range for both cooking temperatures. This implies that, with such a low concentration of pheasant meat, the quantification is not accurate. Overall, the quantification via multiplication factors was effectively applied to the unknown emulsified type sausages. However, as this quantification method normalizes the concentration determined for each species, this method is only practicable when all species added are both known and analyzed together. Therefore, both real-time PCR systems should be expanded if e.g., pork or beef meat were additional ingredients. 

For quantification method C, an additional real-time PCR system was necessary. This system should amplify a specific sequence from all eukaryote species. Therefore, this system is a way to measure the total amount of eukaryotic DNA in a sample. In the literature, many different universal systems have been published. The most common system for quantification of mammal or poultry DNA is the *myostatin* gene. There are several systems which differ in their amplicon length [44,45,46]. However, none of these systems are suited for both of our multiplex real-time PCR systems which detect sequences of the multicopy and mitochondrial *cytochrome b* gene, while *myostatin* is single-copy and nuclear. Another gene which is used quite often is the *18S rRNA* sequence [34,47,48,49]. This gene is multicopy, however, it is also nuclear. Therefore, it was necessary to develop a new real-time PCR system for eukaryotes which amplifies a multicopy and mitochondrial sequence: the *12S rRNA* gene. Because the calculation of the ΔCq is widely used [13], this method was applied in our study. At low cooking temperature, the percentage of the recovery rate within the limits ranged from 50% (guinea fowl meat) to 80% (pheasant meat), and at high cooking temperature, the values were between 73% (guinea fowl meat) and 87% (quail meat). Under both conditions, the CV and bias values were especially large for the lower concentrations. This method allowed the detection of meat from all five poultry species. As an advantage of this quantification system, the amplification of a reference sequence serves also as an inhibition control. However, the addition of another real-time PCR system duplicates the number of samples necessary, and consequently also the costs. Moreover, this quantification is not always precise.

In summary, each quantification method was successfully applied to detect meat from the five species in poultry meat products. While method A had a simple and easy line of action (just a standard curve from standard emulsified sausages), the other two methods were more labor-intensive. For method B, the multiplication factors had to be determined additionally, and for method C, an additional real-time PCR system had to be established and performed. For highly processed food products, an inhibition control is recommended and already included in method C. If the detection system is to be used more often, quantification method B was the easiest to operate: in future experiments, the standard emulsified type sausages are not needed anymore, and the DNA from raw meat can be used for preparing standard curves. In addition, with some minor exceptions, the percentage of acceptable values for CV and recovery rate were the highest for method B.

## 5. Conclusions

Overall, splitting the pentaplex real-time PCR system into one triplex and one duplex real-time PCR system led to a stable, precise, and specific detection method to identify chicken, guinea fowl, pheasant, quail, and turkey meat. All three quantification methods were successfully applied, although mitochondrial gene sequences were chosen. While each quantification method had its pros and cons, a final choice of the quantification method depends on the purpose of its application and the expected concentration of poultry meat species in the meat product.

## Figures and Tables

**Figure 1 foods-09-01049-f001:**
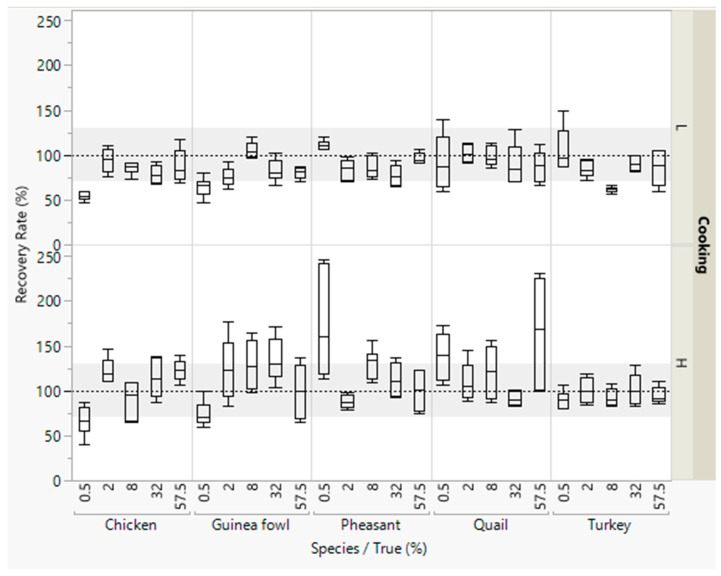
Recovery rates of meat from five poultry species in emulsified type sausages (with 0.5–57.5% meat) quantified with reference material from standard emulsified type sausages (with 0–69% meat). All concentration levels were cooked at low (L) or high temperature (H). DNA was isolated in duplicate from each sausage from both batches, and three independent real-time PCRs were performed, i.e., box plots are from twelve measurements. The grey areas represent the accepted range of 70–130%.

**Figure 2 foods-09-01049-f002:**
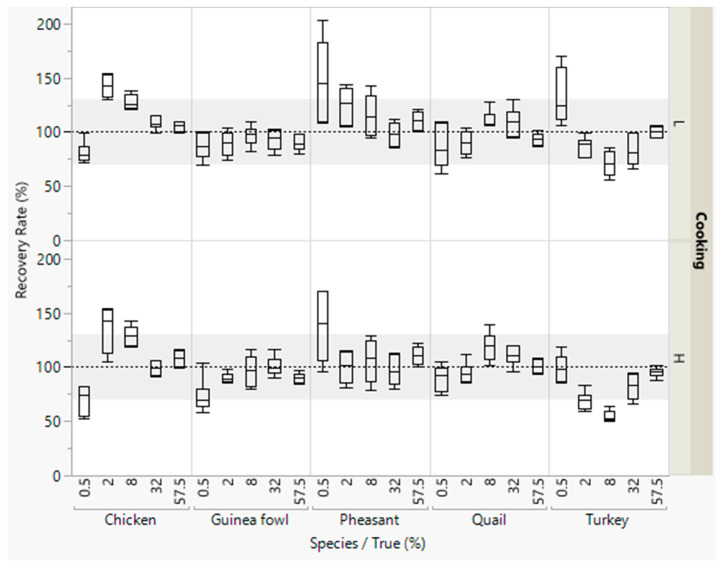
Recovery rates of meat from five poultry species in emulsified type sausages (with 0.5–57.5% meat) quantified with pre-defined multiplication factors. All concentration levels were cooked at low (L) or high temperature (H). DNA was isolated in duplicate from each sausage from both batches, and three independent real-time PCRs were performed, i.e., box plots are from twelve measurements. The grey areas represent the accepted range of 70–130%.

**Figure 3 foods-09-01049-f003:**
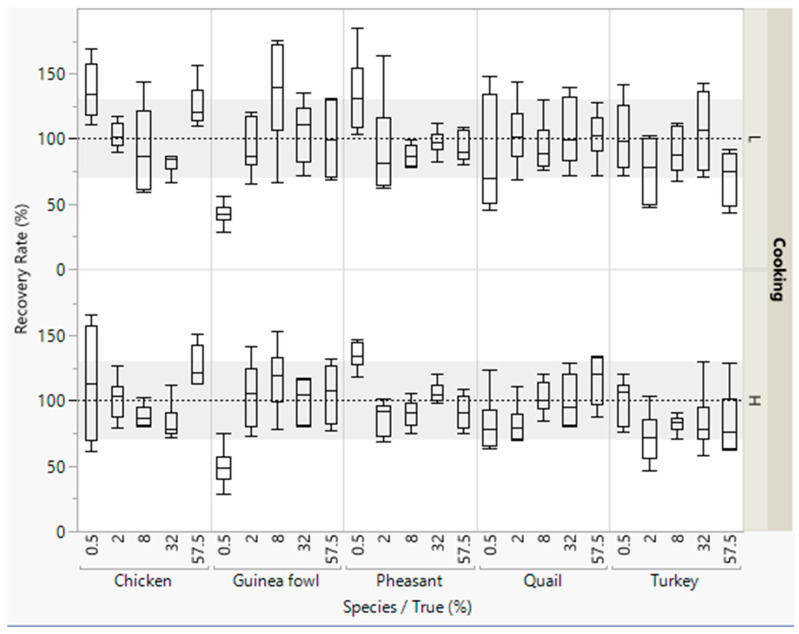
Recovery rates of meat from five poultry species in emulsified type sausages (with 0.5–57.5% meat) quantified with an internal reference sequence. All concentration levels were cooked at low (L) or high temperature (H). DNA was isolated in duplicate from each sausage from both batches, and three independent real-time PCRs were performed, i.e., box plots are from twelve measurements. The grey areas represent the accepted range of 70–130%.

**Table 1 foods-09-01049-t001:** Composition of standard (S1-5) and unknown emulsified type sausages (U1-5).

Amount of Meat Added (%)										
Poultry Species	S1	S2	S3	S4	S5	U1	U2	U3	U4	U5
Chicken	1.0	0.0	69.0	25.0	5.0	2.0	0.5	57.5	32.0	8.0
Guinea fowl	25.0	5.0	1.0	0.0	69.0	32.0	8.0	2.0	0.5	57.5
Pheasant	0.0	69.0	25.0	5.0	1.0	0.5	57.5	32.0	8.0	2.0
Quail	5.0	1.0	0.0	69.0	25.0	8.0	2.0	0.5	57.5	32.0
Turkey	69.0	25.0	5.0	1.0	0.0	57.5	32.0	8.0	2.0	0.5

**Table 2 foods-09-01049-t002:** Sequences of primers and probes.

Multiplex Real-Time PCR	Animal Species	Gene	Code	DNA Sequence 5′–3′	Concentration (µM)	Reference
C-G-P	Chicken	*Cyt b*	C-for	AGC AAT TCC CTA CAT TGG ACA CA	0.20	[27]
			C-rev	GAT GAT AGT AAT ACC TGC GAT TGC A	0.20	
			C-probe	JOE-CAG TCG ACA ACC CAA CCC TTA CCC GAT TC-BHQ1	0.08	[32]
	Guinea fowl	*Cyt b*	G-for	GCA TAC GCC ATC CTC CGC TC	0.20	[33]
			G-rev	GCT GCC CAC TCA GGT TAG A	0.20	
			G-probe	DY682-TGG AGG CGT ACT AGC ACT AGC AGC CTC CG-BHQ2	0.08	[32]
	Pheasant	*Cyt b*	P-for	GAG ACA TGA AAC ACT GGA G	0.20	[33]
			P-rev	CAG GTC CAT TCT ACC AAG G	0.20	
			P-probe	ATTO633-CGT CCT ACT CCT CAC ACT CAT AGC AAC C-BHQ2	0.08	[32]
Q-T	Quail	*Cyt b*	Q-for	TGT ACC CTA CAT CGG CCA AAC C	0.20	[33]
			Q-rev	GTC AGA TGA GAT TCC TAA TGG G	0.20	
			Q-probe	FAM-CCT ACC CTA ACC CGA TTC TTC GCC CTC C-BHQ1	0.10	[32]
	Turkey	*Cyt b*	T-for	CAC TCT TGC ATT CTC TTC TGT GG	0.20	[33]
			T-rev	GGA GGT TAT GGA GGA GTC AAC	0.20	
			T-probe	ROX-CCT ACA CAT GCC GAA ACG TAC AAT ACG-BHQ2	0.08	[32]
ALL	Eukarya	*12S rRNA*	12S-for	AAA CTG GGA TTA GAT ACC CCA CTA TG	0.3	This work
			12S-rev	AGA ACA GGC TCC TCT AGG TGG	0.3	
			12S-probe	FAM-AGA ACT ACG AGC ACA AAC GCT TAA AAC TCT A-BHQ1	0.2	

**Table 3 foods-09-01049-t003:** Cq values for various animal and plant species tested with the triplex real-time polymerase chain reaction (real-time PCR) system (C-G-P) and the duplex real-time PCR system (Q-T) systems.

DNA	Triplex C-G-P						Duplex Q-T			
	Chicken		Guinea Fowl		Pheasant		Quail		Turkey	
Asparagus	32.28	32.96	-	-	-	-	-	-	-	-
Beef	32.20	-	-	-	-	-	-	-	-	-
Chicken	15.21	15.47	-	-	31.27	31.06	-	-	-	-
Deer	-	-	-	-	-	-	34.74	34.26	-	-
Duck	-	-	-	-	34.64	-	-	-	-	-
Goose	-	-	-	-	29.54	29.22	-	-	34.23	-
Guinea fowl	32.95	34.77	16.40	16.71	-	-	-	-	-	-
Kangaroo	-	-	-	-	-	-	32.33	31.46	-	-
Mace	32.13	-	-	-	29.53	-	-	-	-	-
Ostrich	-	-	31.01	30.59	30.49	30.12	30.94	31.20	-	-
Pheasant	-	-	-	-	14.70	14.73	-	-	-	-
Quail	-	-	-	-	-	-	17.69	17.73	-	-
Turkey	-	-	-	-	-	-	-	-	15.86	16.00
Wild boar	-	-	-	-	-	-	-	-	34.80	-
Blank value	32.66	-	32.36	32.33	29.13	29.73	32.23	31.35	31.71	32.38

All samples were measured in duplicates. - no Cq values were obtained until cycle 35.

**Table 4 foods-09-01049-t004:** Predicted concentrations of meat from five poultry species in unknown emulsified type sausages under two temperature conditions, quantified with reference material.

Actual (% *w/w*)	Low Temperature				High Temperature			
	Mean Predicted (% *w/w*) ^a^	SD ^b^	CV (%) ^c^	Bias (%) ^d^	Mean Predicted (% *w/w*) ^a^	SD ^b^	CV (%) ^c^	Bias (%) ^d^
Chicken								
0.5	0.28	0.04	14.41	−43.33	0.33	0.08	24.49	−33.33
2.0	1.88	0.26	14.01	−5.83	2.43	0.27	11.23	21.67
8.0	6.82	0.54	7.95	−14.79	7.22	1.64	22.71	−9.79
32.0	25.22	3.14	12.47	−21.20	36.45	7.15	19.61	13.91
57.5	50.53	10.56	20.90	−12.12	70.88	6.79	9.57	23.28
Guinea fowl								
0.5	0.30	0.06	21.08	−40.00	0.37	0.08	22.27	−26.67
2.0	1.52	0.24	15.83	−24.17	2.48	0.69	27.95	24.17
8.0	8.45	0.73	8.62	5.63	10.30	2.42	23.48	28.75
32.0	26.63	3.93	14.74	−16.77	43.10	8.06	18.69	34.69
57.5	45.90	3.68	8.01	−20.17	57.42	17.53	30.53	−0.14
Pheasant								
0.5	0.55	0.05	9.96	10.00	0.87	0.30	34.74	73.33
2.0	1.68	0.22	13.24	−15.83	1.78	0.15	8.25	−10.83
8.0	6.92	0.96	13.94	−13.54	10.45	1.35	12.96	30.63
32.0	24.85	3.93	15.81	−22.34	36.02	6.30	17.49	12.55
57.5	55.53	3.66	6.59	−3.42	57.60	13.94	24.19	0.17
Quail								
0.5	0.47	0.16	34.99	−6.67	0.70	0.15	22.13	40.00
2.0	2.03	0.20	9.67	1.67	2.22	0.43	19.44	10.83
8.0	7.90	0.88	11.12	−1.25	9.68	2.41	24.91	21.04
32.0	28.83	7.56	26.23	−9.90	29.17	2.35	8.05	−8.85
57.5	50.32	10.41	20.69	−12.49	95.07	37.50	39.45	65.33
Turkey								
0.5	0.52	0.16	31.01	3.33	0.45	0.05	12.17	−10.00
2.0	1.68	0.19	11.53	−15.83	2.03	0.30	14.81	1.67
8.0	4.95	0.23	4.56	−38.13	7.42	0.80	10.83	−7.29
32.0	29.05	2.81	9.68	−9.22	32.60	5.69	17.45	1.88
57.5	49.42	12.04	24.37	−14.06	54.68	5.37	9.82	−4.90

^a^ Values are the means of replicate assays (*n* = 12); ^b^ SD—standard deviation; ^c^ CV—coefficient of variation; ^d^ Bias = 100 * ((mean value − actual value)/actual value).

**Table 5 foods-09-01049-t005:** Matrix-specific multiplication factors to predict the concentration of meat from five poultry species in unknown emulsified type sausages under two temperature conditions.

Temperature	Batch	Chicken	Guinea Fowl	Pheasant	Quail	Turkey
Low	A	1.16	1.19	0.68	3.02	1.19
	B	1.44	1.49	1.12	4.61	1.17
	**Mean**	**1.30**	**1.34**	**0.90**	**3.82**	**1.18**
High	A	0.24	0.12	0.09	0.31	0.61
	B	0.23	0.09	0.10	0.29	0.42
	**Mean**	**0.24**	**0.11**	**0.09**	**0.30**	**0.52**

**Table 6 foods-09-01049-t006:** Predicted concentrations of meat from five poultry species in unknown emulsified type sausages under two temperature conditions, quantified with matrix-specific multiplication factors.

Actual (% *w/w*)	Low Temperature				High Temperature			
	Mean Predicted (% *w/w*) ^a^	SD ^b^	CV (%) ^c^	Bias (%) ^d^	Mean Predicted (% *w/w*) ^a^	SD ^b^	CV (%) ^c^	Bias (%) ^d^
Chicken								
0.5	0.40	0.05	11.70	−19.33	0.35	0.07	19.07	−29.67
2.0	2.85	0.20	6.94	42.58	2.71	0.41	15.24	35.42
8.0	10.23	0.55	5.39	27.83	10.33	0.74	7.13	29.10
32.0	34.77	1.90	5.47	8.65	31.76	2.37	7.47	−0.74
57.5	60.68	2.49	4.10	5.52	62.27	4.51	7.25	8.30
Guinea fowl								
0.5	0.44	0.06	14.08	−13.00	0.37	0.08	21.66	−26.67
2.0	1.80	0.23	12.61	−10.25	1.81	0.09	4.71	−9.58
8.0	7.75	0.73	9.45	−3.10	7.73	1.11	14.42	−3.35
32.0	29.86	2.96	9.91	−6.68	32.29	2.84	8.81	0.90
57.5	51.62	4.15	8.04	−10.22	51.76	2.72	5.26	−9.98
Pheasant								
0.5	0.74	0.19	25.55	47.33	0.69	0.17	24.82	38.00
2.0	2.50	0.33	13.21	24.75	2.00	0.28	14.08	0.00
8.0	9.25	1.60	17.33	15.65	8.50	1.64	19.33	6.19
32.0	31.41	3.44	10.96	−1.83	31.11	4.83	15.53	−2.79
57.5	63.42	5.17	8.16	10.30	63.70	5.12	8.03	10.79
Quail								
0.5	0.43	0.10	22.58	−13.67	0.45	0.06	13.19	−10.33
2.0	1.81	0.22	11.95	−9.75	1.90	0.19	9.94	−5.17
8.0	8.90	0.70	7.83	11.21	9.54	1.06	11.10	19.23
32.0	35.00	4.17	11.93	9.36	35.42	2.81	7.95	10.68
57.5	53.81	3.19	5.93	−6.42	57.99	4.04	6.96	0.86
Turkey								
0.5	0.66	0.13	19.06	32.33	0.50	0.06	12.78	−0.67
2.0	1.74	0.17	9.89	−13.00	1.38	0.16	11.85	−31.00
8.0	5.68	0.94	16.46	−28.98	4.37	0.42	9.65	−45.42
32.0	26.62	4.58	17.19	−16.81	26.32	4.03	15.32	−17.76
57.5	57.65	2.79	4.84	0.26	54.78	2.57	4.69	−4.74

^a^ Values are the means of replicate assay (*n* = 12); ^b^ SD—standard deviation; ^c^ CV—coefficient of variation; ^d^ Bias = 100 * ((mean value—actual value)/actual value).

**Table 7 foods-09-01049-t007:** Predicted concentrations of meat from five poultry species in unknown emulsified type sausages under two temperature conditions, quantified with an internal reference sequence.

Actual (% *w/w*)	Low Temperature				High Temperature			
	Mean Predicted (% *w/w*) ^a^	SD ^b^	CV (%) ^c^	Bias (%) ^d^	Mean Predicted (% *w/w*) ^a^	SD ^b^	CV (%) ^c^	Bias (%) ^d^
Chicken								
0.5	0.69	0.11	16.37	37.67	0.57	0.21	37.23	14.00
2.0	2.06	0.20	9.65	3.08	2.03	0.32	15.90	1.58
8.0	7.38	2.64	35.76	−7.79	7.08	0.68	9.56	−11.56
32.0	26.22	2.48	9.47	−18.05	26.61	4.78	17.97	−16.84
57.5	72.45	9.83	13.56	25.99	73.04	9.12	12.49	27.02
Guinea fowl								
0.5	0.21	0.04	21.00	−57.67	0.25	0.07	30.08	−50.67
2.0	1.87	0.42	22.56	−6.50	2.09	0.50	23.74	4.75
8.0	10.85	3.17	29.24	35.56	9.36	1.99	21.25	17.02
32.0	33.86	7.46	22.03	5.82	32.24	5.27	16.36	0.73
57.5	57.65	17.42	30.21	0.26	60.87	13.56	22.28	5.85
Pheasant								
0.5	0.67	0.15	21.82	34.67	0.67	0.05	7.32	34.67
2.0	1.85	0.76	41.35	−7.50	1.75	0.25	14.53	−12.58
8.0	6.99	0.66	9.46	−12.62	7.22	0.85	11.76	−9.79
32.0	31.17	3.02	9.68	−2.59	34.09	2.49	7.31	6.52
57.5	53.74	6.71	12.49	−6.54	52.72	8.00	15.18	−8.31
Quail								
0.5	0.35	0.16	47.44	−31.00	0.41	0.11	26.76	−18.00
2.0	1.92	0.57	29.44	−3.92	1.65	0.30	18.42	−17.50
8.0	7.33	1.64	22.44	−8.40	8.21	1.02	12.38	2.67
32.0	31.10	5.97	19.20	−2.83	32.02	6.33	19.76	0.06
57.5	57.61	8.82	15.31	0.19	66.62	10.82	16.24	15.87
Turkey								
0.5	0.52	0.14	26.07	4.33	0.50	0.09	17.35	0.33
2.0	1.62	0.51	31.38	−18.92	1.45	0.39	27.19	−27.67
8.0	7.33	1.34	18.28	−8.42	6.59	0.51	7.73	−17.63
32.0	34.91	8.85	25.36	9.10	26.89	7.77	28.90	−15.97
57.5	42.73	9.91	23.19	−25.68	47.90	14.35	29.96	−16.70

^a^ Values are the means of replicate assay (*n* = 12); ^b^ SD—standard deviation; ^c^ CV—coefficient of variation; ^d^ Bias = 100 * ((mean value—actual value)/actual value).

**Table 8 foods-09-01049-t008:** Overview of the applied quantification methods, with percentages of accepted bias, coefficients of variation (CV), and recovery rate.

Method	Technical Summary	Bias	CV	Recovery Rate	Species
		(% within Accepted Range) ^d^			
A ^a^	Quantification with reference material				
	- Fast - Low costs	80	100	68	Chicken
		60	80	65	Guinea fowl
		80	90	80	Pheasant
		80	70	77	Quail
		90	90	85	Turkey
B ^b^	Quantification with matrix-specific multiplication factors				
	- More time and more costs for establishment of multiplication-factors - Suited for repeated use	50	100	70	Chicken
		90	100	93	Guinea fowl
		80	90	82	Pheasant
		100	100	97	Quail
		60	100	73	Turkey
C ^c^	Quantification with an internal reference sequence				
	- More time and more costs due to second real-time PCR assay - With inhibition control	70	80	77	Chicken
		70	70	62	Guinea fowl
		80	90	80	Pheasant
		90	70	80	Quail
		80	40	77	Turkey

^a^ Isolation of DNA from reference material and unknown sample; one real-time PCR for calculation of unknown sample. ^b^ Isolation of DNA from unknown sample, reference material, and raw meat; one real-time PCR for calculation of multiplication factor and another one for calculation of unknown sample. ^c^ Isolation of DNA from reference material and unknown sample; two real-time PCR assays (one for target and one for reference sequence) for calculation of unknown sample. ^d^ for explanation see Section 3.3: Quantification.

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
