# Peer review of "Comparison of Real-Time PCR Quantification Methods in the Identification of Poultry Species in Meat Products"

_foods, 2020, doi:10.3390/foods9081049_

Round 1

Reviewer 1 Report

This is a nice piece of work that focusing on molecular techniques that are considered appropriate for genetic origin authentication.

My suggestion is:

- draft a text of two to three paragraphs to briefly describe other techniques that could be used, providing a comparison with molecular techniques on expected advantages and difficulties.

An example is elemental metabolomics based on diverse signatures such as rare earths.

This is a path to advance the field of genetic origin determination for foods.

Author Response

Reviewer 1

This is a nice piece of work that focusing on molecular techniques that are considered appropriate for genetic origin authentication.

My suggestion is:

- draft a text of two to three paragraphs to briefly describe other techniques that could be used, providing a comparison with molecular techniques on expected advantages and difficulties.

An example is elemental metabolomics based on diverse signatures such as rare earths.

This is a path to advance the field of genetic origin determination for foods.

> Thank you for your nice comments. The intention of this manuscript is to compare available quantification methods for real-time PCR. If the intention would have been to compare available methods for food authenticity we would have been happy to mention elemental metabolomics based on diverse signatures and we will consider this if we will write a review.

Reviewer 2 Report

General remark:

well written, nice piece of work. different quant methods were presented marvelous. however, chapter 3 is a bit long due to doubling of information as text and as tables. Text could be reduced to main issues.

101: sample material

How did you prove the authenticity of the material (is the quail really a quail)? Did you perform an alternative method to check? Please add the chosen procedure?

140: Change off into of

140: combined duplicates

remark: Combining extraction duplicates for further experiments is understandable but will help to hold down "bias", in other words the two independent results from samples are not part of the calculations.

144: dilution

What kind of "dilutor" was used? background-DNA? Please indicate in the text.

220: cut off / blank values

I am bit surprised about the constant blank values around Cq 30 for all systems. Which concentration is used for the PCR? (200 ng in one PCR-reaction?). To you have an explanation for that beside a high DNA conc.?

221: Table 3

Hard to read the table; no clear separation of results between duplex and triplex. Cq duplicates should be illustrated as e.g. n1 and n2 in a header. What does "-" mean? No Cq value after 35 cycles or not tested? Please indicate.

385: "The bioinformatic testing of the primers resulted in no relevant hits as single systems as well as within their combinations (C-G-P and Q-T)."

Sentence should be modified. "No relevant hits" is only valid for species other than C-G-P-Q-T

Author Response

Reviewer 2

General remark:

well written, nice piece of work. different quant methods were presented marvelous. however, chapter 3 is a bit long due to doubling of information as text and as tables. Text could be reduced to main issues.

101: sample material

How did you prove the authenticity of the material (is the quail really a quail)? Did you perform an alternative method to check? Please add the chosen procedure?

> Guinea fowl, pheasant, and quail were bought from a breeder as whole carcasses and were dissected in our technical center. Additionally, a colleague used the same meat samples to establish a LC-MS based technique for differentiating poultry species. This manuscript will be published soon.

140: Change off into of

> done

140: combined duplicates

remark: Combining extraction duplicates for further experiments is understandable but will help to hold down "bias", in other words the two independent results from samples are not part of the calculations.

> Thank you very much for clarifying this aspect. We will consider this point in the design of further experiments.

144: dilution

What kind of "dilutor" was used? background-DNA? Please indicate in the text.

> We used elution buffer as “dilutor”.

220: cut off / blank values

I am bit surprised about the constant blank values around Cq 30 for all systems. Which concentration is used for the PCR? (200 ng in one PCR-reaction?). To you have an explanation for that beside a high DNA conc.?

> In this experiment, the concentration of all animal and plant DNA samples were 2 ng/µl and we used 5 µl for each reaction set-up. Therefore, the DNA-concentration in each reaction was 10 ng and high DNA-concentration is no explanation for these blank values. In each multiplex set up, we have some higher ΔG values and hence, some unspecific reactions are possible.

221: Table 3

Hard to read the table; no clear separation of results between duplex and triplex. Cq duplicates should be illustrated as e.g. n1 and n2 in a header. What does "-" mean? No Cq value after 35 cycles or not tested? Please indicate.

> Thank you for drawing our attention to this difficulty. As you can see, we changed the outline of the table and added the definition of “-“. Each real-time PCR run consisted of 35 cycles and all Cq values until then were noted. If no signal was gained, we declared this as “-“. As the target sequences are mitochondrial and therefore multicopy, we concluded that more cycles were not necessary.

385: "The bioinformatic testing of the primers resulted in no relevant hits as single systems as well as within their combinations (C-G-P and Q-T)."

Sentence should be modified. "No relevant hits" is only valid for species other than C-G-P-Q-T

> Thank you for your advice. We changed “no relevant hits” to “no false positive hits”.

Reviewer 3 Report

Title: I am not sure if the word “ practical” is necessary – it fits more to a guideline rather than to a scientific publication

The reviewed manuscript and presented in it results are very interesting and provide valuable to a wide public knowledge. The manuscript is clear and focused. I think too many time a wording shortcuts have been used. Furthermore, for this kind of a comparison a table joining the summary of the methods and the results of comparison would be useful in my opinion. Otherwise, please see my comments below:

Practical Comparison of Different Quantification Methods for the Detection of Poultry Species in Meat Products – please rephrase as you cannot look for poultry species in meat products – you maybe can identify meat by the poultry species or detect origin of the meat in the meat products

12 detection systems – detecting what? Please be more precise

16 bothersome – is it a necessary word? Please use more scientific word

Abstract: I would expect some values and significance levels of the differences included

28: horsemeat scandal – please provide a reference, this may not be clear for the readers from out of EU or in couple of years when it becomes forgotten

28: It is incorrect to start a sentence from “and” – please rephrase

food labeling – content or ingredients label? To me food label is more a branding of the producing company

where to place the list of ingredients and what it should contain and in which order – I think it is not really important to say that the directive regulates where to place the list of ingredients and in which order - what it should contain is the most important

32: easy – practical?

33: food authenticity - food authenticity testing? Detecting?

35: output – please specify which output, analysis outcomes?

35- 39 please provide literature references

52-54 – please be more clear

57: poultry species in processed meat products as the scope of this manuscript is – I suggest to introduce the mention why will you be focusing on poultry species above that point otherwise please leave it out and let reader wait till line 80 where you introduce the poultry species

58 – 67: could you please elaborate a bit more on the methods?

109: meat species – meat from poultry meat species

94: please be concise and provide the country each time

127: The Wizard DNA isolation kit – please provide the manufacturer

135 please provide the reference

154: with fresh meat as standard curve – with fresh meat based values as standard curve? Please rephrase since now it seems that meat created a standard curve

157: DNA from the poultry meat was used for the standard curves  - was used to prepare/obtain?

174: references needed

181-186 fits more to M&M section, except last sentence of 186

187: what you mean by “all animals”?

188: all poultry - all investigated poultry

There are fairly few of such as mentioned above shortcuts – could you please check the text to avoid those along the manuscript?

189 hits – matches?

192-200 – This fits to me more to M&M section

Results: when the outcomes of the analysis are presented in the table I think there is no need to repeat the exact values in the text.

3.3.4. Comparison – I think it may be beneficial to have a table with the comparisons and describing shortly each quantification method in the first column]

370-374 to me it is more of Introduction

Along the discussion please avoid the repetitions with the introduction and with the results section, as well as M&M section

Author Response

Reviewer 3

Title: I am not sure if the word “ practical” is necessary – it fits more to a guideline rather than to a scientific publication

The reviewed manuscript and presented in it results are very interesting and provide valuable to a wide public knowledge. The manuscript is clear and focused. I think too many time a wording shortcuts have been used. Furthermore, for this kind of a comparison a table joining the summary of the methods and the results of comparison would be useful in my opinion. Otherwise, please see my comments below:

Practical Comparison of Different Quantification Methods for the Detection of Poultry Species in Meat Products – please rephrase as you cannot look for poultry species in meat products – you maybe can identify meat by the poultry species or detect origin of the meat in the meat products

> Thank you very much for drawing our attention to this problem. We changed the title to “Comparison of Quantification Methods in the Identification of Species in Poultry Meat Products”.

12 detection systems – detecting what? Please be more precise

> We substituted the unprecise expression by “analytical methods”.

16 bothersome – is it a necessary word? Please use more scientific word

> Thank your for drawing our attention to this word and apologies for using colloquial language. 

Abstract: I would expect some values and significance levels of the differences included

> We added the lowest and highest value of the recovery rate to the abstract. More details are not possible as the abstract is limited to 200 words and in our opinion, they are not necessary for obtaining a rough idea about the outline of the manuscript.

28: horsemeat scandal – please provide a reference, this may not be clear for the readers from out of EU or in couple of years when it becomes forgotten

> Thank you for your advice. We chose the RASFF report as reference [1].

28: It is incorrect to start a sentence from “and” – please rephrase

food labeling – content or ingredients label? To me food label is more a branding of the producing company

where to place the list of ingredients and what it should contain and in which order – I think it is not really important to say that the directive regulates where to place the list of ingredients and in which order - what it should contain is the most important

> Thank you for your advice. We rephrased and stated more precisely this paragraph.

32: easy – practical?

> done

33: food authenticity - food authenticity testing? Detecting?

> We added the word testing as this describes more clearly the purpose.

35: output – please specify which output, analysis outcomes?

> We changed the word “output” to “analysis results” and hope, that the meaning is more precise now.

35- 39 please provide literature references

> As you can see, we added two references. One is about the reason why it can be necessary to know, if one ingredient is higher than the other one [2]. And the other reference is about why thresholds are important [3].

52-54 – please be more clear

> We added some additional information and hopefully the reason is now clearer why this quantification method is not suitable.

57: poultry species in processed meat products as the scope of this manuscript is – I suggest to introduce the mention why will you be focusing on poultry species above that point otherwise please leave it out and let reader wait till line 80 where you introduce the poultry species

> Thank you for this advice. We decided to wait for mentioning the focus of this manuscript until later to have a better structure of the manuscript.

58 – 67: could you please elaborate a bit more on the methods?

> We explained the two quantification methods more in detail.

109: meat species – meat from poultry meat species

> Thank you for suggestion. We kindly accepted this.

94: please be concise and provide the country each time

> Our apologies, we added the country.

127: The Wizard DNA isolation kit – please provide the manufacturer

> We added the manufacturer; however, all manufacturers are mentioned in section 2.1.1 Chemical material.

135 please provide the reference

> Again apologies, we added the missing information.

154: with fresh meat as standard curve – with fresh meat based values as standard curve? Please rephrase since now it seems that meat created a standard curve

> Thank you for drawing our attention to this headline. We changed it to “Quantification with matrix-specific multiplication factors” and hope that the approach of this quantification method is now clear.

157: DNA from the poultry meat was used for the standard curves  - was used to prepare/obtain?

>Again, we changed the wording according to the remark above.

174: references needed

> We added the companies as well as the city and country.

181-186 fits more to M&M section, except last sentence of 186

> We disagree with your opinion as we think, that this section is part of the results and is essential for all further results.

187: what you mean by “all animals”?

> Excuse our shortcut. With “all animals” we meant “all entries for animal organisms in the database”. 

188: all poultry - all investigated poultry

There are fairly few of such as mentioned above shortcuts – could you please check the text to avoid those along the manuscript?

> Our apologies again. We tried to substitute all shortcuts with more precise words.

189 hits – matches?

> We changed the word “hits” to “matches”.

192-200 – This fits to me more to M&M section

> We disagree with your opinion as we think, that this section is part of the results and are essential for all further results.

Results: when the outcomes of the analysis are presented in the table I think there is no need to repeat the exact values in the text.

> We agree with your opinion. Therefore, we only mentioned the lowest and highest values, or the mean with SD. However, we think for a good comparison those values are worthwhile to mention.

3.3.4. Comparison – I think it may be beneficial to have a table with the comparisons and describing shortly each quantification method in the first column]

> Thank you for your suggestion. As you can see, we added an additional table (Table 8) describing the workflow and the percentual values which were within the range for bias, CV, and recovery rate.

370-374 to me it is more of Introduction

> We agree with you and moved this part to the introduction.

Along the discussion please avoid the repetitions with the introduction and with the results section, as well as M&M section

> Thank you for drawing our attention to this problem. We deleted as much as possible of the duplications to keep the understanding.

References:

  1. Commission, E. The rapid alert system for food and feed 2013 annual report. 2013.
  2. Köppel, R.; Ganeshan, A.; van Velsen, F.; Weber, S.; Schmid, J.; Graf, C.; Hochegger, R. Digital duplex versus real-time PCR for the determination of meat proportions from sausages containing pork and beef. European Food Research and Technology 2018, 245, 151-157, doi:10.1007/s00217-018-3147-8.
  3. Kesmen, Z.; Yetiman, A.E.; Sahin, F.; Yetim, H. Detection of chicken and turkey meat in meat mixtures by using real-time PCR assays. J Food Sci 2012, 77, C167-173, doi:10.1111/j.1750-3841.2011.02536.x.

Reviewer 4 Report

I have to admit that though the subject was interesting research with enough data generated. However, I suggest the authors should address the following minor comments for further consideration.

Line no. 37 to 40.: What authors meant the ingredient x and y in Introduction section?

Line no. 44 – suggested to use ‘RT-PCR’. Entire manuscript as well.

Try to avoid oral English language style. Eg. Line no. 80, 368 – we focused, ‘we compared’.

  • The manuscript should be proofread by native speaker for the correction of English language so as to meet the standard of publication in the journal.
  • Abbreviations must be defined at first mention and used consistently thereafter in entire manuscript. Authors must revise entire manuscript.
  • Please write the conclusion again and be more concise to the major findings.

Author Response

Reviewer 4

I have to admit that though the subject was interesting research with enough data generated. However, I suggest the authors should address the following minor comments for further consideration.

Line no. 37 to 40.: What authors meant the ingredient x and y in Introduction section?

> Thank you for this question. The “x” and “y” are variables which stand representative for all possible ingredients. As you can see, we rewrite this paragraph.

Line no. 44 – suggested to use ‘RT-PCR’. Entire manuscript as well.

> Thank you for this advice. However, we tried to implement the MIQE guidelines to our work and “RT-qPCR” is advised as the abbreviation for “reverse transcription-qPCR”. Therefore, we agree with those authors, that the abbreviation “RT-PCR” for real-time PCR is confusing [1]. To avoid confusion, we decided to use “real-time PCR” throughout the manuscript.

Try to avoid oral English language style. Eg. Line no. 80, 368 – we focused, ‘we compared’.

> We apologies for using colloquial language. If possible, we transferred all “we” sentences into passive form.

The manuscript should be proofread by native speaker for the correction of English language so as to meet the standard of publication in the journal.

> Thank you for drawing our attention to language problems. We kindly asked a colleague for assistant in proof reading.

Abbreviations must be defined at first mention and used consistently thereafter in entire manuscript. Authors must revise entire manuscript.

> Thank you for your advice. We revised the entire manuscript and explained all abbreviations at their first mention.

Please write the conclusion again and be more concise to the major findings.

> As you can see, we revised the conclusions.

References:

  1. Bustin, S.A.; Benes, V.; Garson, J.A.; Hellemans, J.; Huggett, J.; Kubista, M.; Mueller, R.; Nolan, T.; Pfaffl, M.W.; Shipley, G.L., et al. The MIQE guidelines: minimum information for publication of quantitative real-time PCR experiments. Clin Chem 2009, 55, 611-622, doi:10.1373/clinchem.2008.112797.